## 1 Widespread stratospheric intrusion influence on summer ozone pollution over China revealed

- by multi-site ozonesondes and validated EAC4 reanalysis
- 3 Zhiheng Liao<sup>1</sup>, Jinqiang Zhang<sup>2</sup>, Meng Gao<sup>3</sup>, Zhiqiang Ma<sup>1</sup>

- 1 Institute of Urban Meteorology, China Meteorological Administration, Beijing, China
- 2 State Key Laboratory of Atmospheric Environment and Extreme Meteorology, Institute of Atmospheric Physics,
- Chinese Academy of Sciences, Beijing, China
- 3 Department of Geography, Hong Kong Baptist University, Hong Kong SAR, China

Correspondence: Z. Ma (zqma@ium.cn)

#### Abstract

Understanding stratospheric intrusion (SI) is crucial for elucidating atmospheric complexities and improving strategies to mitigate surface ozone (O<sub>3</sub>) pollution. This study investigates a deep trough-induced SI event in China from June 10 to 13, 2013, based on ozonesondes from Beijing, Changchun, and Hong Kong, and validated O<sub>3</sub> reanalysis products. Ozonesondes from Beijing indicated notable upper-level secondary O<sub>3</sub> peaks (> 400 ppbv) since June 11. Tropospheric sub-high O<sub>3</sub> layers were observed in Changchun on June 12 (> 120 ppbv) and Hong Kong on June 13 (> 80 ppbv). Nationwide surface measurements recorded severe O<sub>3</sub> pollution (> 100 ppbv) from western plateaus to eastern plains over China. Together, these observations suggest a widespread influence of stratospheric O<sub>3</sub> intrusion. Further, the ozonesonde-validated EAC4 reanalysis reproduced the fine-scale SI structure (O<sub>3</sub>-rich "tongue"), in turn well explaining the secondary O<sub>3</sub> peaks and sub-high O<sub>3</sub> layers in ozonesonde observations. The O<sub>3</sub>-rich "tongue" swept through the Tibetan Plateau on June 10, triggering extreme O<sub>3</sub> pollution with a stratospheric contribution up to 30 ppbv (>30 %). With the trough's eastward movement, the O<sub>3</sub>-rich "tongue" penetrated into the lower troposphere of eastern China, and then entrained into the surface layer, exacerbating surface O<sub>3</sub> pollution occurred in eastern China on June 13, with a stratospheric O<sub>3</sub> contribution of 3–15 ppbv (2–10 %). This research underscores the importance of multi-site ozonesondes in understanding stratospheric O<sub>3</sub> intrusions and the potential of the publicly available EAC4 reanalysis in multiyear SI analyses.

**Keywords**: stratospheric intrusion; secondary ozone peak; surface ozone pollution; upper-level trough; contribution

## 1 Introduction

Surface ozone  $(O_3)$  poses significant risks to public health and ecosystem productivity due to its strong oxidative capacity (Monks et al., 2015). While  $O_3$  in the lower atmosphere is predominantly produced through photochemical reactions, stratospheric intrusion (SI)—the process where  $O_3$ -rich air masses from the stratosphere descend to the lower troposphere—can also increase surface  $O_3$  concentrations in certain regions (Akritidis et al., 2018;Skerlak et al., 2019;Dreessen, 2019). The natural SI processes complicate efforts to manage and reduce anthropogenic  $O_3$  pollution (Zhao et al., 2025). Therefore, quantifying the SI contribution to surface  $O_3$  is essential for developing effective air quality management strategies.

 SI is a key component of extratropical weather processes, and detecting SI events along with their influence on tropospheric chemistry has been a major scientific focus across Europe (Appenzeller and Davies, 1992;Stohl et al., 2003;Akritidis et al., 2018), North America (Hocking et al., 2007;Lin et al., 2016;Wang et al., 2020b), East Asia (Lin et al., 2021;Liu et al., 2024;Chen et al., 2024), and other extratropical regions (Zhang et al., 2024). While observational evidence confirms that SI can trigger episodic spikes in surface O<sub>3</sub> concentrations (Cristofanelli et al.,

2010; Langford et al., 2012; Yates et al., 2013; Lin et al., 2015; Dreessen, 2019; Ou-Yang et al., 2022; Chen et al., 2023; Chen et al., 2024), accurately quantifying this phenomenon remains difficult due to observational limitations. Traditionally, ozonesondes have served as a primary tool for SI detection, as they provide complete vertical O<sub>3</sub> profiles up to approximately 35 km. However, their sparse temporal and spatial coverage hinders a comprehensive understanding of how stratospheric O<sub>3</sub> penetrates to the surface (Chen et al., 2011; Zhao et al., 2021; Hong et al., 2024). To date, much of the current understanding of SI and its role in surface O<sub>3</sub> pollution has been derived from satellite observations (Li et al., 2015; Zhang et al., 2022; Jaeglé et al., 2017), atmospheric reanalysis (Chen et al., 2023; Knowland et al., 2017; Bartusek et al., 2023; Akritidis et al., 2018), and model simulations (Wang et al., 2020a;Zhao et al., 2021;Zhang et al., 2022;Chang et al., 2023;Hong et al., 2024;Luo et al., 2024;Zhao et al., 2024; Zhu et al., 2024; Skerlak et al., 2019). Yet, these approaches often lack rigorous validation against ozonesonde data, leading to substantial uncertainties in their findings. Alternatively, some studies have attempted to quantify stratospheric influences using ground-based chemical tracers, e.g., the ratio of O<sub>3</sub> to CO (O<sub>3</sub>/CO) (Ma et al., 2014; Chen et al., 2024), cosmogenic sulfur-35 (35S) (Lin et al., 2016; Lin et al., 2021), and the ratio of cosmogenic beryllium-10 to beryllium-7 (10Be/1Be) (Jordan et al., 2003;Liu et al., 2024). However, these methods also exhibit limitations, as ground-based measurements alone cannot fully resolve the vertical structure of SI aloft (Zheng et al., 2024). Moreover, inconsistencies arise when different tracers are applied—for instance, while one study using <sup>35</sup>S (Lin et al., 2021) suggested a west-high-east-low SI contribution pattern over China, another study based on the O<sub>2</sub>/CO ratio (Chen et al., 2024) reported the opposite distribution. These conflicting results reveal a significant research gap, emphasizing the necessity for comprehensive analysis of multi-platform datasets to advance our understanding of stratospheric O<sub>3</sub> intrusion and its effects on surface O<sub>3</sub> pollution.

This study examines a representative SI event triggered by an upper-level trough over China during 10-13 June 2013. The event was characterized by severe surface  $O_3$  pollution that sequentially occurred in the high-altitude Tibetan Plateau and low-lying eastern China. To investigate the potential connection between SI processes and these  $O_3$  pollution episodes, we employed a multi-platform integrated approach combining multi-site ozonesondes, ground-based  $O_3$  measurements, satellite  $O_3$  products, and atmospheric  $O_3$  reanalysis. Through comprehensive analysis of this multi-platform datasets, we aim to address two key scientific objectives: 1) to characterize the spatiotemporal evolution of trough-induced stratospheric  $O_3$  intrusion, 2) to quantitatively assess the SI contribution to surface  $O_3$  pollution over different regions.

## 2 Datasets

### 2.1 Ozonesonde observation

In China, ozonesondes, along with radiosondes, were routinely launched weekly in Beijing ( $116.47 \,^{\circ}$ E,  $39.80 \,^{\circ}$ N) and Hong Kong ( $114.17 \,^{\circ}$ E,  $22.31 \,^{\circ}$ N). During June 2013, an intensive ozonesonde experiment was held in Beijing and Changchun ( $125.20 \,^{\circ}$ E,  $43.90 \,^{\circ}$ N), with consecutive launches from June 10-13. The details of the ozonesonde experiment can be found in Zhang et al. (2013). These sondes (including the routine ozonesonde in Hong Kong) were launched around 13:30 China Standard Time, providing high-resolution profiles of  $O_3$  partial pressure, atmospheric pressure, temperature, and humidity from the surface up to approximately 35 km (Zhang et al., 2021;Liao et al., 2024). For this study, data from nine ozonesonde observations were analyzed to examine stratospheric  $O_3$  intrusion during June 10-13, 2013, including four consecutive days in Beijing and Changchun, and a single launch on June 13 in Hong Kong. By comparing the sonde-based surface  $O_3$  concentrations with ground-based  $O_3$  measurements (Fig. 3B), we demonstrated a good accuracy of these ozonesonde observations (R = 0.981, and RAB = 3.2 ppbv).

### 2.2 Atmospheric reanalysis data

ERA5, the fifth-generation ECMWF (European Centre for Medium-Range Weather Forecasts) global reanalysis, offers a comprehensive dataset at a spatial resolution of  $0.25^{\circ} \times 0.25^{\circ}$  and a temporal resolution of 1 hour for climate and weather analysis (Hersbach et al., 2020). It integrates model data with observations using four-dimensional variational assimilation in ECMWF's Integrated Forecast System (IFS). This study utilized ERA5 data, including geopotential height, potential vorticity, and wind fields, to describe the synoptic conditions during the stratospheric intrusion event.

EAC4 (ECMWF Atmospheric Composition Reanalysis 4) represents the fourth generation of ECMWF's atmospheric composition reanalysis, with a spatial resolution of  $0.75 \,^{\circ} \times 0.75 \,^{\circ}$  and a temporal resolution of 3 hours (Inness et al., 2019). EAC4 assimilates data from various satellite sources, including total column  $O_3$  from the Ozone Monitoring Instrument and Global Ozone Monitoring Experiment-2 on Metop satellites, profile data from the Microwave Limb Sounder, and partial columns from Solar Backscatter Ultra-Violet and Ozone Mapping and Profiler Suite. Note that surface  $O_3$  measurements and ozonesonde  $O_3$  profile data in China are not assimilated into the EAC4 reanalysis. The IFS used in EAC4 incorporates an extended version of the Carbon Bond 2005 chemical mechanism, which includes 126 tropospheric reactions. The emission datasets are composed of anthropogenic emissions from the MACCity inventory (Granier et al., 2011), biogenic emissions from MEGAN2.1 model (Guenther et al., 2006), and biomass burning emissions from the Global Fire Assimilation System (Kaiser et al., 2012). Apart from  $O_3$ , the stratospheric  $O_3$  tracer ( $O_3S$ ,  $O_3$  originating from the stratosphere) is also provided in EAC4 reanalysis. This study employed both  $O_3$  and  $O_3S$  to characterize the three-dimensional structure of stratospheric  $O_3$  intrusion.

### 2.3 Auxiliary data

- Additional data sources included ground-based O<sub>3</sub> measurements from the China National Air Quality Monitoring
  Network and the Hong Kong Environmental Protection Department, satellite cloud images from the Moderate

  Passelvtion Imagina Spectromaliameter (MODIS) satellite O<sub>2</sub> meadwate from the Atmospheric Infrared Soundary
- Resolution Imaging Spectroradiometer (MODIS), satellite O<sub>3</sub> products from the Atmospheric Infrared Sounder (AIRS) (Aumann et al., 2003), and atmospheric O<sub>3</sub> reanalysis from the Modern-Era Retrospective Analysis for
- (AIRS) (Aumann et al., 2003), and atmospheric O<sub>3</sub> reanalysis from the Modern-Era Retrospective Analysis for Research and Applications, Version 2 (MERRA2) (Gelaro et al., 2017). According to previous studies (Jaegléet al.,
- Research and Applications, Version 2 (WERRA2) (Gelato et al., 2017). According to previous studies (Jaegreet al., 2017; Knowland et al., 2017; Zhang et al., 2022), we used satellite O<sub>3</sub> retrieved from AIRS Level 3 product, which
- has a spatial resolution of  $1^{\circ} \times 1^{\circ}$ . In contrast, the MERRA2 reanalysis has a spatial resolution of  $0.5^{\circ} \times 0.625^{\circ}$ .
- Both AIRS and MERRA2 O<sub>3</sub> products served as alternative references to EAC4 O<sub>3</sub> reanalysis to provide a
- large-scale view of horizontal and vertical O<sub>3</sub> structures during the SI event. Hourly surface O<sub>3</sub> concentrations from
- 77 cities in China (including Hong Kong) were used to assess nationwide O<sub>3</sub> pollution during the stratospheric
- intrusion event.

## 3 Results

#### 3.1 Ozonesonde evidence of stratospheric O<sub>3</sub> intrusion

Fig. 1 illustrates the evolution of the upper-level trough event from June 10 to 13, 2013. On June 10, the upper-level trough extended from the Mongolian Plateau towards the Tibetan Plateau. By June 11, the trough had moved eastward and deepened into a "V-shaped" structure between 90°E and 120°E, causing an extremely distorted westerly jet and strong northerlies at the western flank of the trough. On this day, the emerged 1.5 PVU potential vorticity contours at 400 hPa provide convincing evidence for a deep stratospheric intrusion. On June 12, the "V-shaped" trough persisted at 200 hPa. By June 13, the upper-level trough had weakened to be a shallow structure over the North China Plain (NCP). Three-dimensional dynamics associated with upper-level troughs

involves stratospheric dry intrusion (SDI) and warm conveyor belts (WCB) airstreams (Browning and Roberts, 1994;Browning, 1997). The SDI originates in the lower stratosphere on the cold side of the trough (west of the trough axis) and descends behind the cold front, while the WCB originates in the warm sector of the trough (east of the trough axis), ascending rapidly to the mid- and upper troposphere. During this event, these contrasting airstreams led to significantly different weather conditions at the two sides of the trough, with cloudy weather in the WCB zone (east) and clear weather in the SDI zone (west). There appeared an obvious transition from cloudy to clear weather in the eastern China with the eastward movement of upper-level trough. On June 13, China, excluding the northeast and eastern coastal regions, experienced clear weather.

**Fig. 1**. (A) Horizontal distribution of geopotential height (shading, units in gpm), and wind direction of jet stream in excess of 20 m s<sup>-1</sup> (arrows) at 200 hPa, and potential vorticity of 1.5 PVU (blue contours) at 400 hPa. (B)

MODIS satellite cloud images with the dashed box marking eastern China (105 E–121 E, 21 N–41 N). Red dot lines in (A) denote the axis of upper-level trough at 200 hPa. Magenta circles mark the available ozonesondes at different sites (BJ: Beijing, CC: Changchun, and HK: Hong Kong) on different days.

146147148

153154

158

Previous ozonesonde-based observational studies (Lemoine, 2004; Hwang et al., 2007; Chen et al., 2011; Ojha et al., 2017) revealed that secondary O<sub>3</sub> peak in a height range between 9 and 16 km (i.e., near the tropopause) is a characteristic O<sub>3</sub>-profile structure when SI occurs and triggers tropopause folding. The continuous and multi-site ozonesondes in this study provided a unique opportunity to characterize stratospheric O<sub>3</sub> intrusion linked to upper-level trough from an observational perspective (Fig. 2). On June 10, before the trough arrived, Beijing was influenced by WCB airstreams, showing high relative humidity (> 60%) in the upper troposphere. By June 11, Beijing was near the trough axis, and the O<sub>3</sub>-rich SDI airstream began to affect the upper atmosphere, creating a secondary O<sub>3</sub> peak (~400 ppbv at 9.5 km height) just above the rapidly descended thermal tropopause (which dropped from 10.5 km on June 10 to 8.2 km on June 11). Besides, the cold dry air of the SDI lead to a quick drop in relative humidity from 70% on June 10 to below 25 % on June 11 in the upper troposphere of Beijing. On June 12 and 13, the secondary O<sub>3</sub> peaks continued to be observed over Beijing, with peak concentrations rising to 650 ppbv by June 13, but the altitude of these peaks gradually increased up to 13.6 km by June 13 with the increase of thermal tropopause height. Unlike that in Beijing, the sonde-based O<sub>3</sub> profiles in Changchun showed secondary O<sub>3</sub> peak only in June 13, when upper-level trough moved eastward to affect Changchun. However, sub-high O<sub>3</sub> layer (> 120 ppbv) appeared in the middle troposphere (4.2–8.1 km height, the shaded light gray in Fig. 2) in advance on June 12, accompanied by extremely low relative humidity. This sub-high O<sub>3</sub> layer is likely the transport result of pre-intruded O<sub>3</sub> from stratosphere over Beijing or its surroundings. Similar sub-high O<sub>3</sub> layer (> 80 ppbv) also occurred in the lower troposphere (3.5-6.0 km height, the shaded light gray in Fig. 2) of Hong Kong (a subtropical city) on June 13. These high-O<sub>3</sub> and low-humidity air masses in the troposphere reflect obvious stratospheric origin, suggesting a widespread SI influence from extratropics to subtropics during this deep trough event.

Fig. 2.  $O_3$  vertical distribution over (A) Beijing, (B) Changchun, and (C) Hong Kong derived from ozonesonde and other data sources (including AIRS satellite observation, EAC4 and MERRA2 reanalysis) during 10–13 June 2013. Black and blue lines denote the sonde-based temperature (T) and relative humidity (RH) profiles, respectively. Gray dashed lines represent the thermal tropopause height, and gray dot lines indicate the boundary layer top height. Upper-level secondary  $O_3$  peaks are shaded heavy gray, and SI-induced  $O_3$ -rich layer in the troposphere is shaded light gray.

## 3.2 Three-dimensional structure of stratospheric O<sub>3</sub> intrusion

The multi-site ozonesonde observations only provide a snapshot of stratospheric  $O_3$  intrusion. To further visualize its three-dimensional structure, we introduced the commonly-used  $O_3$  products, including AIRS satellite

observation, MERRA2 and EAC4 reanalysis (Li et al., 2015;Knowland et al., 2017;Akritidis et al., 2018). These three large-scale  $O_3$  products were firstly validated against our ozonesonde observations. As shown in Fig. 2, AIRS satellite observation missed the upper-level secondary  $O_3$  peaks and the boundary-layer  $O_3$  enhancements. MERRA2 reanalysis captured the secondary  $O_3$  peaks but still showed large negative biases to the observed boundary-layer  $O_3$  enhancements. In contrast, EAC4 reanalysis reproduced well the major features of the  $O_3$  vertical distribution, including upper-level secondary  $O_3$  peaks and boundary-layer  $O_3$  enhancements. Particularly, EAC4 exactly captured the SI-induced sub-high  $O_3$  layers in the middle troposphere of Changchun (on June 12) and the lower troposphere of Hong Kong (on June 13). This qualitative comparison suggests that EAC4 had a powerful ability to reproduce both the SI dynamics and boundary-layer photochemical processes. The scatter comparison with quantitative statistics in Fig. 3A further demonstrate that EAC4  $O_3$  reanalysis had the strongest correlation (R = 0.947), the lowest mean absolute bias (MAB = 19.1 ppbv), the lowest root mean square error (RMSE = 36.9 ppbv), and the largest index of agreement (IOA = 0.985) to the ozonesonde observation. This sonde-based validation (Fig. 3A), along with validation against with nationwide ground-based  $O_3$  observations (Fig. 3B and Fig. 5A), provides us enough confidence in adopting EAC4 reanalysis to explore the three-dimensional structure of trough-induced stratospheric  $O_3$  intrusion.

**Fig. 3**. (A) Validation of AIRS, MERRA2, and EAC4  $O_3$  products with 9 ozonesonde observations from Beijing, Changchun, and Hong Kong. (B) Validation of EAC4/Sonde-based surface  $O_3$  concentrations with ground-based  $O_3$  observations. In (A), AIRS, MERRA2, and EAC4  $O_3$  data were spatially interpolated to the location of ozonesonde stations. In (B), ground-based  $O_3$  observations across 466 sites in 76 cities were resampled to EAC4 grid (0.75  $^{\circ}$  × 0.75  $^{\circ}$ ) for comparison with EAC4-based surface ozone reanalysis; ground-based  $O_3$  observations at three neighboring sites (Tiantan site in Beijing, Daishan Park site in Changchun, and Sham Shui Po site in Hong Kong) were used for comparison with Sonde-based surface ozone concentrations. *N*, *R*, MAB, RMSE, and IOA denote the number of statistic samples, correlation coefficient, mean absolute bias, root mean square error, and index of agreement, respectively.

Fig. 4 illustrates the EAC4-based three-dimensional structure of upper-level trough-induced stratospheric O<sub>3</sub> intrusions over China. High O<sub>3</sub> concentrations at 200 hPa aligned with the trough location, extending southwestward (June 10) and southward (June 11–13) along the trough axis, which well explained the upper-level secondary O<sub>3</sub> peaks over Beijing since June 11 and over Changchun on June 13 (Fig. 2A and 2B). The stratospheric intrusions developed into elongated (about 2000 km) and slender (about 2000 km) streamers with elevated O<sub>3</sub> concentrations exceeding 150 ppbv (referred to as SDI-induced O<sub>3</sub>-rich belts) at 400 hPa. On the east of the SDI

streamers, the WCB streamers were parallel with anomalously low  $O_3$  concentrations (referred to as WCB-related  $O_3$ -poor belts). On June 12, the SDI-induced  $O_3$ -rich belt stretched to northeastern China, explaining the observed sub-high  $O_3$  layer in the middle troposphere of Changchun (Fig. 2B). In the lower troposphere (700 hPa),  $O_3$ -rich air masses appeared over subtropical southern China on June 11, the strongest SI day, indicating the southern edge of stratospheric  $O_3$  intrusion. This lower-tropospheric  $O_3$ -rich air masses persisted in subsequent days and be exactly captured by Hong Kong's ozonesonde on June 13 (Fig. 2C). From June 11 to 13, there was a significant northeastward transport and dispersion of  $O_3$ -rich filament due to the strengthening southwesterly winds in the lower troposphere of eastern China. Through vertical and horizontal transport, lower tropospheric  $O_3$  concentrations increased by approximately 20 ppbv across eastern China, consistent with the 18 ppbv  $O_3$  increase observed in 2–6 km height over Beijing, indicating widespread enhancement of lower tropospheric  $O_3$  background due to stratospheric  $O_3$  intrusion and accumulation.

222223224

228

231

234235

236237

242243

213214

Compared with total O<sub>3</sub>, O<sub>3</sub>S provides a more direct view of stratospheric intrusion (Fig. 4B). The three-dimensional O<sub>3</sub>S structure depicts the upper-level trough-induced stratospheric O<sub>3</sub> intrusion as a sheet-like lowering of the O<sub>3</sub>S-rich layer along the western flank of the trough and an O<sub>3</sub>S-rich tongue extending southward and westward from the trough base. These features aligned well with the typical structure of extratropical stratospheric intrusion associated with tropopause folding (Bithell et al., 1999; Hocking et al., 2007). On June 10, stratospheric O<sub>3</sub> intrusion directly hit the Tibetan Plateau, triggering extremely high surface O<sub>3</sub> concentrations. From June 11 to 13, the O<sub>3</sub>S-rich tongue progressed eastward into eastern China with the trough's eastward movement. Unlike that in the Tibetan Plateau, the O<sub>3</sub>S-rich tongue in eastern China was blocked in the lower free troposphere and did not further intrude the surface layer. This result agreed well with the observed sub-high O<sub>3</sub> layer in 3.5-6.0 km height over Hong Kong (Fig. 2C), suggestive of no direct stratospheric O<sub>3</sub> intrusion to the surface in the low-elevation eastern China. Nevertheless, these O<sub>3</sub>-rich stratospheric air masses can be further transported into atmospheric boundary layer via convective mixing pathway, contributing to boundary layer O<sub>3</sub> increase. In this process, their stratospheric characteristics (high O<sub>3</sub>, low humidity) tend to be lost due to strong turbulence mixing, eventually becoming unrecognizable in atmospheric boundary layer. Interestingly, another stratospheric intrusion induced by severe tropical storm (name: "Yagi") over the Northwest Pacific provided a parallel reference (Fig. 4B). Compared with the tropical storm-induced stratospheric O<sub>3</sub> instruction, the upper-level trough-induced intrusion descended to a relatively lower altitude, causing widespread O<sub>3</sub>S signals in the atmospheric boundary layer over eastern China. Note that apart from the trough- and storm-induced SI, a secondary SI emerged in the upwind of upper-level trough on June 12 likely driven by peripheral compensatory flows. This secondary SI lead to elevated O<sub>3</sub> concentrations over the Mongolian Plateau (Fig. 4A). However, they were not further transported into eastern China.

Fig. 4. (A) Spatial distribution of  $O_3$  concentrations in 200, 400, 700 hPa, and surface layer. (B) Three-dimensional structure of  $O_3S$  concentrations. In (A), white dashed lines mark the trough axis at 200 hPa, and magenta half-circles mark the locations of ozonesondes at different sites (Beijing, Changchun and Hong Kong) on different days.

## 3.3 Stratospheric intrusion contribution to surface O<sub>3</sub> pollution

245246

247248

250251

254255

256257

261262

264265

267268

Fig. 5A presents the spatial distribution of surface O<sub>3</sub> concentrations derived from ground-based measurements and EAC4 reanalysis during the SI event. The EAC4-based surface O<sub>3</sub> reanalysis agreed well with nationwide ground-based observations (R = 0.697, MAB = 12.4 ppbv, RMSE = 23.5 ppbv, and IOA = 0.961, Fig. 3B), confirming the reliability of the EAC4 reanalysis again as the previous validation with ozonesondes. On June 10, the Tibetan Plateau experienced high O<sub>3</sub> concentrations near or exceeding 80 ppbv, with observed O<sub>3</sub> in Lhasa reaching up to 100 ppbv at 14:00 BJT. In contrast, eastern China exhibited low O<sub>3</sub> concentrations (< 40 ppbv) due to cloudy and rainy weather on this day. From June 10 to 13, surface O<sub>3</sub> concentrations decreased day by day in the Tibetan Plateau, while they increased from west to east in eastern China. By June 13, eastern China suffered severe O<sub>3</sub> pollution, with observed O<sub>3</sub> concentrations exceeding 100 ppbv in most of the NCP cities. From June 10 to 13, the continuous stratospheric dry intrusion led to a weather transition from cloudy to cloudless in eastern China (Fig. 1B), enhancing photochemical  $O_3$  production due to the abundance of  $O_3$  precursors over there. On the other hand, strong solar radiation in cloudless weather promoted the development of thermal convection, facilitating the mixing of pre-intruded O<sub>3</sub>-rich stratospheric air from the lower free troposphere into the surface layer. These two mechanisms combined to trigger severe O<sub>3</sub> pollution in eastern China on June 13. The continue ozonesondes in Beijing provide convincing evidence for these two mechanisms. Returning to Fig. 2A, boundary-layer O<sub>3</sub> concentrations in Beijing increased significantly from 57.8 ppbv on June 10 to 120.6 ppbv on June 13. Considering the sharp O<sub>3</sub> gradient in the interface between the atmospheric boundary layer and the lower free troposphere, the dramatic increase in boundary-layer O<sub>3</sub> can be primarily attributed to photochemical production (Liao et al., 2024). However, the concurrent rise in O<sub>3</sub> concentrations in the lower free troposphere (an 18 ppbv O<sub>3</sub> increase in 2–6 km height from June 10 to 13) indicated that stratospheric O<sub>3</sub> intrusion contributed to elevating lower-tropospheric O<sub>3</sub> background, ultimately exacerbating boundary layer O<sub>3</sub> pollution.

Fig. 5. (A) Surface spatial distribution of total  $O_3$  concentration derived from ground-based measurement (dots) and EAC4 reanalysis (shading). (B) Surface spatial distribution of  $O_3S$  concentration derived from EAC4 reanalysis. (C) Spatial distribution of  $O_3S$  fraction in surface  $O_3$  concentration calculated from EAC4 reanalysis.

To quantify the contribution of stratospheric intrusion to surface O<sub>3</sub> pollution, Fig. 5B illustrates the spatial distribution of EAC4-based surface O<sub>3</sub>S concentrations during the SI event, and Fig. 5C shows the contribution fraction (CF) of  $O_3S$  in surface  $O_3$  concentrations (CF =  $100\% \times O_3S/O_3$ ). The high-elevation Tibetan Plateau received high-concentration O<sub>3</sub> from stratospheric intrusion, particularly on June 10 when the upper-level trough was oriented northeast-southwest towards the Tibetan Plateau. On this day, surface O<sub>3</sub>S cocnentration exceeded 30 ppbv (up to 48.5 ppbv) in the Tibetan Plateau, contributing to over 30 % of the surface O<sub>3</sub> concentration (up to 44.7 %). Subsequent days saw a gradual decrease in O<sub>3</sub>S over the Tibetan Plateau. On June 12 and 13, significant O<sub>3</sub>S hotspots (> 20 ppbv) appeared in the Mongolian Plateau. In conjunction with the three-dimensional O<sub>3</sub>S structure (Fig. 4B), the elevated O<sub>3</sub>S concentrations in the Mongolian Plateau can be attributed to the emerged secondary SI on June 12 rather than initial trough-induced SI. It seems that the elevated O<sub>3</sub>S in the Mongolian Plateau had no influences on surface O<sub>3</sub> over eastern China considering its downwind location in the lower troposphere. Nonetheless, eastern China was affected not only by the "fresh" stratospheric air in the eastward-movement O<sub>3</sub>S-rich tongue (via convective mixing), but also by the pre-intruded "aged" stratospheric air from the Tibetan Plateau (via eastward transport). Due to continuous accumulation, region-averaged O<sub>3</sub>S concentrations increased approximately 1.0 ppbv in eastern China from June 10 to 13, whereas their fraction in surface O<sub>3</sub> decreased from 11.8 % to 8.3 % as local O<sub>3</sub> photochemical production accelerated. On June 13, surface O<sub>3</sub>S concentrations in eastern China ranged from 3 to 15 ppby, accounting for 2–10 % of surface O<sub>3</sub> concentrations. In the highly polluted NCP region, O<sub>3</sub>S contributed approximately 10 % of surface O<sub>3</sub>, reflecting a nonnegligible role of stratospheric  $O_3$  intrusion in exacerbating  $O_3$  pollution.

### **4 Conclusions and Discussion**

276277278

279280

283

288

290

295

This study reveals that the upper-level trough-induced stratospheric O<sub>3</sub> intrusion over China did not occur as a local-scale vertical descent from the stratosphere to the lower troposphere just at the mid-latitude location where tropopause folding occurs; instead, it involved a long-range transport from mid-latitude tropopause folding zone (e.g., Beijing) to lower-latitude areas (e.g., Hong Kong), featuring an O<sub>3</sub>-rich "tongue" structure with upper-level secondary O<sub>3</sub> peak at the base of tongue (e.g., over Beijing) and lower-tropospheric sub-high O<sub>3</sub> layer at the tip of tongue (e.g., over Hong Kong). The O<sub>3</sub>-rich "tongue" swept through the high-elevation Tibetan Plateau when the upper-level trough extended towards this highland region at its initial stage, triggering extreme surface O<sub>3</sub> pollution. With the eastward movement of upper-level trough, the O<sub>3</sub>-rich "tongue" penetrated into the lower troposphere of low-elevation eastern China. Over there, the intruded O<sub>3</sub>-rich stratospheric air masses in the lower troposphere, including the "fresh" stratospheric air vertically transported from O<sub>3</sub>-rich "tongue" and the "aged" stratospheric air horizontally transported from the Tibetan Plateau, were then entrained into the atmospheric boundary layer via lower-tropospheric dynamic processes (e.g. convective mixing). At the same time, the strengthening lower-tropospheric southwesterly winds with the eastward movement of upper-level trough gradually participated to transport these O<sub>3</sub>-rich stratospheric air back to the mid-latitudes, ultimately exacerbating surface O<sub>3</sub> pollution in the NCP region (e.g., Beijing). While several SI events have been reported in China (Chang et al., 2023; Hong et al., 2024;Li et al., 2015;Luo et al., 2024;Wang et al., 2020a;Zhang et al., 2022;Zhao et al., 2024), this trough-induced SI episode may be the first event of its widespread impact and refined structure documented (Fig. 6).

316 317

319

324

332

302

306

308

The quantitative stratospheric intrusion contributions derived from the validated EAC4 reanalysis are generally consistent with previous model results in China. In the low-elevation eastern China, surface O<sub>3</sub>S concentrations were previously estimated to be in the range of 5-20 ppbv during the SI events (Wang et al., 2020a; Zhang et al., 2022; Chang et al., 2023). Our EAC4-based estimation agreed well with this range, reflecting the typical magnitude of SI contribution in the low-elevation eastern China. As for the high-elevation Tibetan Plateau, a case-based model study (Skerlak et al., 2019) revealed that stratospheric tracer concentrations at the surface reach peak values of 20 % of the imposed stratospheric value, and a month-based model study (Yin et al., 2023) suggested that 36.5 % of surface O<sub>3</sub> in the hotspot of the southern Tibetan Plateau was contributed by stratospheric O<sub>3</sub> intrusion. Our EAC4-based estimation was comparable to these fractional contributions, corroborating the potential of SI to significantly influence surface O<sub>3</sub> concentrations in this highland region. Besides, ground-based chemical tracer method had been developed to quantify the stratospheric intrusion contribution over China. While Chen et al. (2024) identified the nationwide SI-induced O<sub>3</sub> enhancement as a west-low-east-high spatial distribution pattern based on surface O<sub>3</sub> and CO observations, Lin et al. (2021) determined a west-high-east-low spatial distribution pattern of SI-induced O<sub>3</sub> contribution based on ground-based cosmogenic <sup>35</sup>S observations at the Himalayas and beyond. Our result appears to support the latter, which conforms to the common knowledge that the highland regions are more susceptible to stratospheric intrusion because of their proximity to the stratosphere (Skerlak et al., 2019; Wang et al., 2020b;Lin et al., 2021).

Fig. 5. Schematic illustration of upper-level trough-induced stratospheric  $O_3$  intrusion influence on surface  $O_3$  pollution over China

To the best of our knowledge, this study is the first to utilize continuous and multi-site ozonesondes to investigate stratospheric O<sub>3</sub> intrusion. While we acknowledge that a single case study may not be fully representative, it effectively demonstrates the value of continuous and multi-site ozonesonde measurements in enhancing our understanding of stratospheric O<sub>3</sub> intrusion phenomena. On the other hand, these continuous and multi-site ozonesondes provide a valuable and unique benchmark for examining the capacity of those commonly-used O<sub>3</sub> products (including AIRS satellite observation, MERRA2 and EAC4 O<sub>3</sub> reanalysis) in characterizing stratospheric O<sub>3</sub> intrusion. Previous study indicated that MERRA2 can be used in scientific studies to identify SIs by both atmospheric dynamics and composition (Knowland et al., 2017). Here, we demonstrate that EAC4, a publicly available dataset from European Centre for Medium-Range Weather Forecasts, performs better than MERRA2 in quantitatively characterizing stratospheric O<sub>3</sub> intrusion via comparative evaluation. Moreover, in contrast to MERRA2, EAC4 simulates full O<sub>3</sub> chemistry in the troposphere (an extended version of the Carbon Bond 2005 (CB05) chemical mechanism), allowing to determine the influence of stratospheric O<sub>3</sub> on surface concentrations separate from photochemically produced O<sub>3</sub>. Therefore, this is a proof opening the door to detailed multiyear analyses of stratospheric O<sub>3</sub> intrusion and their quantitative contribution to surface O<sub>3</sub> over China and worldwide based on the publicly available EAC4 O<sub>3</sub> reanalysis.

Data availability. All used data, excluding the ozonesonde in Beijing and Changchun, are open source. ERA5 atmospheric data are available from Copernicus Climate Change Service (C3S) Climate Data Store accessible at https://cds.climate.copernicus.eu/. EAC4 O<sub>3</sub> reanalysis were obtained from Copernicus Atmospheric Monitoring Service Data Store accessible at https://ads.atmosphere.copernicus.eu/. The MODIS true color images are available from the NASA Earth Observations (NEO): https://neo.gsfc.nasa.gov/. The AIRS and MERRA2 O<sub>3</sub> products were

- obtained from Goddard Earth Sciences Data and Information Services Center accessible at
- https://disc.gsfc.nasa.gov/. The ozonesonde data in Hong Kong were obtained from World Ozone and Ultraviolet
- Radiation Data Centre accessible at https://woudc.org/. The ozonesonde data in Beijing and Changchun are
- available from the first author upon reasonable request (zhliao@ium.cn).

364

### **Author contributions**

- Z.L. conceived the original idea, analyzed the data, and wrote the first version manuscript. J.Z. designed intensive
- ozonesonde experiments in Beijing and Changchun. Z.M supervised the research project. All authors discussed the
- results and commented on the manuscript.

369

# **Competing interests**

The authors declare no competing interests.

371372

#### Acknowledgements

- This research has been supported by the National Natural Science Foundation of China (Grant Nos. 42405115,
- 42293321 and 42207115).

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
