# Peer review of "Widespread stratospheric intrusion influence on summer ozone pollution over China revealed"

_EGUsphere, 2025_

## Author Comment (AC1)

**Reply to Anonymous Referee #2**

Review for Liao et al., "Widespread stratospheric intrusion influence on summer ozone pollution over China revealed by multi-site ozonesonde, ground-based measurement and fully-validated reanalysis"

This study investigates a stratospheric intrusion (SI) event over China from June 10 to 13, 2013, using a combination of ozonesonde measurements, ground-based observations, and reanalysis data. The authors aim to characterize the SI event, quantify its contribution to surface ozone pollution, and elucidate the underlying dynamical transport mechanisms. The research effectively highlights the importance of ozonesonde observations in validating reanalysis data and improving our understanding of SI impacts on ozone pollution.

Overall, this is good study that combines ozonesonde data from multiple locations (Beijing, Changchun, and Hong Kong) with nationwide ground-based measurements and the EAC4 reanalysis product. This multi-pronged approach provides a comprehensive view of the SI event and its impact.

Additionally, the use of ozonesondes provides valuable vertical ozone profile information, which is crucial for characterizing SI events. The ozonesonde data also significantly aids in the evaluation of the EAC4 reanalysis data. While it's not entirely clear if these observations are directly assimilated into EAC4, they do provide important and independent constraints on the reanalysis results, adding confidence to the study's conclusions.

Reply: So glad to receive your positive comments. We have carefully considered your suggestions and comments, and made corresponding modifications and explanations.

Hence, I suggest the following modifications:

The authors should check with the EAC4 team and add information about the assimilation of Chinese ozonesonde data in EAC4. Providing more information about the ozone-related chemistry in EAC4 (e.g., emission of ozone precursor gases, and whether surface observations from China are included in their assimilation) would help the reader.

Reply: Thank you for this constructive comment. Surface $O_3$ measurements and ozonesonde $O_3$ profile data in China are not assimilated into the EAC4 reanalysis. We clarified this in the revised manuscript. The assimilated $O_3$ retrievals in EAC4 consist of multiple satellite data (SCIAMACHY, MIPAS, MLS, OMI, GOME-2 and SBUV/2). The emission datasets in EAC4 are composed of anthropogenic emissions from the MACCity inventory (Granier et al., 2011), biogenic emissions from MEGAN2.1 model (Guenther et al., 2006), and biomass burning emissions from the Global Fire Assimilation System (Kaiser et al., 2012). In the revised manuscript, we added the abovementioned information and highlighted that other details of the EAC4 can be found in Inness et al. (2019). The following Table from Inness et al. (2019) shows the basic information of the EAC4 model system (i.e., CAMSRA).

|  | MACCRA | CIRA | CAMSRA |
| --- | --- | --- | --- |
| Period covered | 2003–2012 | 2003–2018 | 2003–2016 (will be extended) |
| Assimilation system | IFS Cycle 36r1 4D-Var | IFS Cycle 40r2 (2003–2015) 4D-Var
IFS Cycle 41r1 (2016–2018) 4D-Var | IFS Cycle 42r1 4D-Var |
| Horizontal resolution | 80 km globally (T255) | 110 km globally (T159) | 80 km globally (T255) |
| Temporal resolution (output frequency) | 6-hourly analysis fields
3-hourly forecast fields from 00:00 UTC up to 24 h | 6-hourly analysis fields
3-hourly forecast fields from 06:00 and 18:00 UTC up to 12 h | 3-hourly analysis fields
3-hourly forecast fields from 00:00 UTC up to 48 h
1-hourly surface forecast fields from 00:00 UTC up to 48 h |
| Anthropogenic emissions | Chemistry species: MACCity (trend: ACCMIP + RCP8.5), Aerosols: AEROCOM | MACCity (trend: ACCMIP + RCP8.5) and CO emission upgrade from Stein et al. (2014) for chemistry and aerosols | MACCity (trend: ACCMIP + RCP8.5) and CO emission upgrade from Stein et al. (2014) |
| Biomass burning emissions | GFED (2003–2008) and GFASv0 (2009–2012) | GFASv1.2 | GFASv1.2 |
| Biogenic emissions | Monthly mean VOC emissions for the year 2003 calculated by the MEGAN2.1 model (Guenther et al., 2006) used for the whole period. No interannual variability. | Monthly mean VOC emissions calculated by the MEGAN2.1 model (Guenther et al., 2006) using MERRA reanalysed meteorology (Sindelarova et al., 2014) for the period 2003–2010. For the remaining years a climatology of the MEGAN–MACC data was used. | Monthly mean VOC emissions calculated by the MEGAN model using MERRA reanalysed meteorology (Sindelarova et al., 2014) for 2003–2016. |
| Chemistry modules | CTM MOZART3 coupled to the IFS (see Flemming et al., 2009) | IFS(CB05) (Flemming et al., 2015) and Cariolle ozone parametrisation in stratosphere,
CHEM_VER=ver14wd | IFS(CB05) (Flemming et al., 2015, with updates documented in Sect. 2.1.2) and Cariolle ozone parametrisation in stratosphere,
CHEM_VER=ver15 |
| Aerosol modules | Morcrette et al. (2009) | Morcrette et al. (2009) plus changes described in Flemming et al. (2017b) | Morcrette et al. (2009) with changes documented in Sect. 2.1.1. |
| Input meteorological observations | ECMWF NWP (stream = DA) | ECMWF NWP (stream = DCDA) | As in ERA5 (2003–2016) |
| Assimilated $O_3$ retrievals | GOME, SCIAMACHY, MIPAS, MLS, OMI and SBUV/2 | GOME, SCIAMACHY, MIPAS, MLS, OMI, GOME-2 and SBUV/2 | SCIAMACHY, MIPAS, MLS, OMI, GOME-2 and SBUV/2 |
| Assimilated CO retrievals | MOPITT, IASI | MOPITT | MOPITT |
| Assimilated $NO_2$ retrievals | SCIAMACHY | – | SCIAMACHY, OMI and GOME-2 |
| Aerosol used in radiation scheme | Tegen climatology | Tegen climatology | Interactive aerosols, i.e. aerosol fields from CAMSRA used in radiation scheme |
| Ozone used in radiation scheme | GEMS climatology | GEMS climatology (2003–2015) MACCRA climatology (2016–2018) | Interactive ozone, i.e. ozone field from CAMSRA used in radiation scheme |
| Stratospheric chemistry | Yes | No, but Cariolle ozone parametrisation in stratosphere and stratospheric $O_3$ available. | No, but Cariolle ozone parametrisation in stratosphere and stratospheric $O_3$ available. |

Granier, C., Bessagnet, B., Bond, T., D'Angiola, A., Denier van der Gon, H., Frost, G. J., Heil, A., Kaiser, J. W., Kinne, S., Klimont, Z., Kloster, S., Lamarque, J.-F., Liousse, C., Masui, T., Meleux, F., Mieville, A., Ohara, R., Raut, J.-C., Riahi, K., Schultz, M. G., Smith, S. G., Thompson, A., van Aardenne, J., van der Werf, G. R., and van Vuuren, D. P.: Evolution of anthropogenic and biomass burning emissions of air pollutants at global and regional scales during the 1980–2010 period, Climatic Change, 109, 163–190, https://doi.org/10.1007/s10584-011-0154-1, 2011.

Guenther, A., Karl, T., Harley, P., Wiedinmyer, C., Palmer, P. I., and Geron, C.: Estimates of global terrestrial isoprene emissions using MEGAN (Model of Emissions of Gases and Aerosols from Nature), Atmos. Chem. Phys., 6, 3181–3210, https://doi.org/10.5194/acp-6-3181-2006, 2006.

Inness, A., Ades, M., Agusti-Panareda, A., Barré, J., Benedictow, A., Blechschmidt, A. M., Dominguez, J. J., Engelen, R., Eskes, H., Flemming, J., Huijnen, V., Jones, L., Kipling, Z., Massart, S., Parrington, M., Pench, V. H., Razinger, M., Remy, S., Schulz, M., and Suttie, M.: The CAMS reanalysis of atmospheric composition, Atmos Chem Phys, 19, 3515-3556, 10.5194/acp-19-3515-2019, 2019.

Kaiser, J. W., Heil, A., Andreae, M. O., Benedetti, A., Chubarova, N., Jones, L., Morcrette, J.-J., Razinger, M., Schultz, M. G., Suttie, M., and van der Werf, G. R.: Biomass burning emissions estimated with a global fire assimilation system based on observed fire radiative power, Biogeosciences, 9, 527–554, https://doi.org/10.5194/bg-9-527-2012, 2012.

Please clarify the use of AIRS Level 3 data. For these type of studies you should use level 2 data. Add more information detailing what specific product was used, and how was it processed. Also, how sensitive AIRS is to the surface ozone levels.

Reply: Thanks for this comment. We are confused about your suggestion to use the AIRS Level 2 data, rather than the Level 3 data, which is a more advanced version produced by the AIRS Science Team/Joao Texeira (2013). The AIRS Level-3 ozone retrievals have been commonly used to estimate vertical structure of ozone in stratospheric intrusion events (Knowland et al., 2017; Zhang et al., 2022; Jaeglé et al., 2017). We listed these references and added the spatial resolution ($1^o \times 1^o$) information of AIRS in the revised manuscript. Our previous evaluation based on long-term ozonesondes in Beijing (Zhang et al., 2024) indicated that AIRS widely retrieves lower ozone values near the surface, and, upward from 700 hPa, larger values are consistently presented by AIRS in the troposphere below 300 hPa. Additionally, AIRS seems to frequently underestimate the ozone between 300 and 100 hPa; however, at numerous pressure levels above 100 hPa, AIRS tends to detect higher ozone concentrations. In this study, we actually further demonstrated that ARIS ozone product had a relatively lower accuracy when compared to EAC4 and MERRA2 reanalysis.

AIRS Science Team/Joao Texeira (2013). AIRS/Aqua L3 Daily Standard Physical Retrieval (AIRS+AMSU) 1 degree ×1 degree V006. Greenbelt, MD, USA: Goddard Earth Sciences Data and Information Services Center (GES DISC).

Knowland, K. E., Ott, L. E., Duncan, B. N., and Wargan, K.: Stratospheric Intrusion-Influenced Ozone Air Quality Exceedances Investigated in the NASA MERRA-2 Reanalysis, Geophys Res Lett, 44, 10691-10701, 10.1002/2017GL074532, 2017.

Zhang, Y. J., Li, J., Yang, W. Y., Du, H. Y., Tang, X., Ye, Q., Wang, Z. X., Sun, Y. L., Pan, X. L., Zhu, L. L., and Wang, Z. F.: Influences of stratospheric intrusions to high summer surface ozone over a heavily industrialized region in northern China, Environ Res Lett, 17, Artn 094023 10.1088/1748-9326/Ac8b24, 2022.

Jaeglé, L., Wood, R., and Wargan, K.: Multiyear Composite View of Ozone Enhancements and Stratosphere-to-Troposphere Transport in Dry Intrusions of Northern Hemisphere Extratropical Cyclones, Journal of Geophysical Research: Atmospheres, 122, 13,436-413,457, https://doi.org/10.1002/2017JD027656, 2017.

Zhang J., Xuan Y., Bian J., Vömel H., Zeng Y., Bai Z., Li D., Chen H., 2024. Comparison between ozonesonde measurements and satellite retrievals over Beijing, China. Atmospheric and Oceanic Science Letters, 17 (2024), 100378.

If feasible, include a broader discussion about SI events. Placing the June 2013 event in a broader climatological context would be beneficial. Given the availability of the EAC4 reanalysis data, the authors should consider including climatological statistics to illustrate the typical influence of SI events on surface ozone on annual and monthly timescales. This analysis would provide valuable perspective on the typical impact of SI on surface ozone in China and contextualize the significance of the specific 2013 event analyzed in this study.

Reply: Thank you for this excellent comment. To be candid, our initial plan was to incorporate climatic statistical characterization after this case analysis. We preliminarily identified 90 summertime upper-level trough cases from 2003 to 2022 using a subjective identification method, and each case persisted 1-7 days. However, we faced a large challenge in composite analysis of these identified cases due to the irregular trough structure. We thought that the "reversed Ω-shaped" trough-induced SI cases can be compositely characterized through a cyclone-centric composite method proposed by Jaeglé et al. (2017). However, the "V-shaped" trough-induced SI cases can not be composited through the cyclone-centric composite method (no cyclonic center can be found for some cases). Beside, as shown in the 2013 case, the locations of deep trough determine the SI influencing areas over the China (Tibetan Plateau or Eastern Plain). Due to the extremely contrasting altitudes between the western and eastern China, the magnitude of SI contribution is also very different between these two areas, implying that the composite analysis should take into account of the locations of SI to avoid the substantial smooth of stratospheric influences at the difference surface areas. The above challenges have forced us to temporarily abandon our initial plan. Still, the climatological statistical characterization is on our schedule, but it needs more time and effort to address the abovementioned challenges (mainly in composite analysis of "V-shaped" trough cases). Once we complete the climatological statistical characterization, we will write another paper to present the results.

Given that this reply will be openly presented in the interactive discussion process, here we show three additional SI cases (2005-08-03, 2019-08-15, and 2022-06-14) in the following figure to enhance the significance of the specific 2013 event analyzed in this study.

[Figure]

Fig. R1 The identified upper-level trough days in summer of 2003-2022 and three cases of trough-induced stratospheric intrusion.

Jaeglé, L., Wood, R., and Wargan, K.: Multiyear Composite View of Ozone Enhancements and Stratosphere-to-Troposphere Transport in Dry Intrusions of Northern Hemisphere Extratropical Cyclones, Journal of Geophysical Research: Atmospheres, 122, 13,436-413,457, https://doi.org/10.1002/2017JD027656, 2017.

If the authors can modify the manuscript accordingly, I recommend it for publication.

Reply: Thanks for your excellent comments again. We have carefully considered your suggestions and comments, and made corresponding modifications and explanations. Regarding to the climatological statistical characterization, we explain the reasons why it was not presented in this study.

---

## Author Comment (AC2)

**Reply to Anonymous Referee #1**

This study combined ozonesonde observations, validated EAC4 reanalysis products, and ground-level measurement data to characterize an ozone stratospheric intrusion (SI) event in June 2013 in 3 cities of China and claimed to have developed a method for quantifying the contribution of SI events to ground-level ozone pollution based on EAC4 reanalysis data. This study is innovative and representative in the field of atmospheric environmental chemistry and atmospheric science as so far, but the following possible doubts remained. I recommend the publication of the manuscript after the authors have properly answered or resolved these possible questions below.

Reply: Thank you very much for your positive and detailed comments. We have carefully considered your suggestions and comments, and made corresponding modifications and explanations.

**Major Comment:**

1. **Lines 150-151:** "High-level secondary ozone peaks are a characteristic $O_3$-profile structure associated with tropospheric folding, a major form of SI in the extratropical region" How do you define the range of high-level secondary ozone peaks? It didn't seem to be specific values but a range. Why it is a major form of SI? Here maybe need a simple explanation by 1-2 sentences.

   Reply:Thank you for this comment. Secondary $O_3$ peak is defined as the abnormal $O_3$ peak near the tropopause (9-16 km). Here, "secondary" is relative to the first $O_3$ peak in the stratosphere. We clarified this definition in the revised manuscript. Referring to previous ozonesonde-based observational studies, we claimed that secondary $O_3$ peak in a height range between 9 and 16 km (i.e., near the tropopause) is a characteristic $O_3$-profile structure when SI occurs and triggers tropopause folding.    Lines 148-150.

2. It's true that Beijing and Changchun are belong to extratropical region, but I'm not sure if Hong Kong is. Maybe it's more subtropical or tropical. If so, can the SI still be determined based on the secondary ozone peaks?

   Reply:Thank you for pointing it out. Hong Kong is a subtropical city. In Hong Kong, we observed a sub-high $O_3$ layer in the lower free troposphere (4.2–8.1 km height), rather than the upper-level $O_3$ peak near the tropopause. The sub-high $O_3$ layer in Hong Kong is a transport result of the extratropical stratospheric $O_3$ intrusion. However, secondary $O_3$ peaks in the extratropical region represent the initial and major characteristic of stratospheric intrusion.

3. There are some confusions in **Table 1**: Which statistical comparisons were based on Beijing, Changchun and Hong Kong, and which were based on the 76 cities? This should be described more clearly in the table. In addition, it would be best to give in the table the number of data points N used for correlation analysis, which is an important statistical parameter. Finally, how about the correlation between $O_3$-Sonde & $O_3$-Ground? Maybe it has been shown in Zhang et al. 2013, but it's better to be analyzed here by the same statistic method in this study to tell readers the accuracy of ground-level ozone measurements by ozonesondes. In addition to Table 1, I suggest the authors also include scatter plots in the supplement to compare the observation data and ozone products.

Reply: Thank you for picking it up. According to your comments and suggestion, we present the statistical comparisons in the form of scatter plots in this revision, instead of the Table form in the initial manuscript. In the scatter plots, we list the important statistical parameters, including the data points $N$, correlation coefficient $R$, mean absolute bias MAB, root mean square error RMSE, and index of agreement IOA. Besides, we add the correlation statistic between the $O_3$-Sonde & $O_3$-Ground using 9 ozonesonde samples in three cities (Beijing, Changchun, and Hong Kong). Although a very limited ozonesonde samples, the $O_3$-Sonde correlated well with the $O_3$-Ground (N=9, R=0.981, MAB=3.2 ppbv, RMSE=9.8 ppbv, and IOA=0.994), indicating the accuracy of ozonesondes. Lines 85-87 and Figure 3.

[Figure]

Figure 3. (A) Validation of AIRS, MERRA2, and EAC4 $O_3$ products with 9 ozonesonde observations from Beijing, Changchun, and Hong Kong. (B) Validation of EAC4/Sonde-based surface $O_3$ concentrations with ground-based $O_3$ observations. In (A), AIRS, MERRA2, and EAC4 $O_3$ data were spatially interpolated to the location of ozonesonde stations. In (B), ground-based $O_3$ observations across 466 sites in 76 cities were resampled to EAC4 grid ($0.75° × 0.75°$) for comparison with EAC4-based surface ozone renalaysis; ground-based $O_3$ observations at three neighboring sites (Tiantan site in Beijing, Daishan Park site in Changchun, and Sham Shui Po site in Hong Kong) were used for comparison with Sonde-based surface ozone concentrations. $N$, $R$, MAB, RMSE, and IOA denote the number of statistic samples, correlation coefficient, mean absolute bias, root mean square error, and index of agreement, respectively.

4. **Lines 237-239:** Please include a scatter plot in the supplement for comparing observation data and EAC4 ozone reanalysis product.

   Reply: Thanks for this suggestion. As you suggested, we embedded a scatter plot in Figure 3 to compare observation data and other ozone products. Seeing Figure 3 in the revised manuscript.

5. **Section 3.3.** How did you get the contribution of stratospheric intrusions to surface ozone pollution? More specifically, how did you quantitatively distinguish between local photochemical production and stratospheric intrusion? Should there be a description of the relevant calculation process or equations in Section 2 (*Datasets*)?

Reply: Thanks for your suggestions. The ECA4 provide two ozone indicators, the total ozone ($O_3$) and the stratospheric ozone tracer ($O_3S$). The stratospheric ozone tracer is a part of total ozone, representing the stratospheric ozone contribution in total ozone. So the contribution fraction is calculated as $CF = 100\% \times O_3S/O_3$. Line 280.

**Minor Comment:**

1. **Lines 43-45:** When citing multiple studies, could you also specify where the SI events occurred?

   Reply: Thanks for this suggestion. In the revised manuscript, we specified where the SI events occurred. (Lines 40-43).

2. **Lines 58-60:** Any other literate from places other than China?

   Reply:Regarding the opposite conclusions drawn from different chemical tracers, no other literate had been reported from places other than China.

3. **Lines 76-78:** Move the sentence of "Ozonesondes provide … " to Introduction paragraph 2. Just describe the method you used here, with the ozonesondes parameter shown in detail, and if it is refered to Zhang et al. 2013 please please specify this point.

   Reply: Thanks for this suggestion. We moved the mentioned sentence to Introduction paragraph 2 (Lines 46-48). Besides, we specified that the details of the intensive ozonesonde experiment can be found in Zhang et al. (2013). (Lines 79-80)

4. **Lines 155-156:** "Similar sub-high ozone layer (> 80 ppbv) also occurred in the lower troposphere (3.5–6.0 km height) of Hong Kong on June 13." What were the similarities? Was the extreme low humidity and the sub-high ozone layer occurring at the same time? But the humidity in Hong Kong was not extremely low. Why there was no sub-high ozone layer in Beijing and Changchun on June 13 (4.0–6.0 km height also with high-ozone and low-humidity condition)?

   Reply: Thanks for this comment. The sub-high $O_3$ layer emerged in the lower troposphere (3.5–6.0 km height) of Hong Kong on June 13 is similar to that occurred in the middle troposphere (4.2–8.1 km height) of Changchun on June 12. Both of them are featured with extreme low humidity and high ozone concentration. The relative humidity in Hong Kong declined to 20%, reaching an extreme low value in subtropical coastal region. Although this humidity is not as low as that in Changchun (near zero), but it is lowered from the relatively higher humidity background (80% below and above the sub-high $O_3$ layer). For a subtropical coastal city such as Hong Kong, its humidity background is significantly higher than that in Changchun (a middle-latitude inland city). In fact, the declining amplitudes of RH in the sub-high $O_3$ layer are similar in Hong Kong (from 80% declined to 20%) and Changchun (from 60% declined to 0%).

   Why there was no sub-high ozone layer in Beijing and Changchun on June 13? This is a good question. Three-dimensional dynamics associated with upper-level troughs involves stratospheric

dry intrusion (SDI) and warm conveyor belts (WCB) airstreams. The SDI originates in the lower stratosphere on the cold side of the trough (west of the trough axis) and descends behind the cold front, while the WCB originates in the warm sector of the trough (east of the trough axis), ascending rapidly to the mid- and upper troposphere. Given the dynamical characteristics of upper-level trough, the mid-latitude region is mainly controlled by SDI airstream (the dynamical mechanism of upper-level secondary $O_3$ peaks over Beijing and Changchun). When WCB airstream entangles part of the dry-intrusion air masses in the relatively lower latitudes, it will lead to northward recirculation of pre-intruded $O_3$ to mid-latitude region, causing the occurrence of sub-high $O_3$ layer in the middle troposphere (the dynamical mechanism of sub-high $O_3$ layer over Changchun on June 12). On June 13, both Beijing and Changchun was far away from WCB airstream due to its eastward movement, so no occurrence of sub-high ozone layer.

**Lines 155-158:** The author observed a sub-high ozone layer in the lower troposphere of Hong Kong on June 13. The vertical profiles of T, RH, and $O_3$ at that layer are like those around 4 km in the troposphere of Beijing on June 10$^{th}$. But there is no discussion for the latter one.

Reply: On June 10$^{th}$, the $O_3$ concentration showed no peak around 4 km above Beijing (despite slight increase around 4 km but peak around 8 km) although the relative humidity presented a valley. This structure is not similar to that (the co-occurrence of $O_3$ peak and RH valley) in Changchun (on June 12$^{th}$) and Hong Kong (on June 13$^{th}$). We had no enough evidence to attribute the low-humidity layer over Beijing to stratospheric intrusion. So, there is no discussion for this low-humidity layer.

5. **Line 194:** I can not see the trough axis in Fig.3. Could the author highlight them in the figure?

   Reply: Thanks for this suggestion. We added the trough axis in the revised manuscript. Figure 4A.

6. **Line 197:** I don't how 2000 km is concluded here.

   Reply: Thank you for picking it up. We labeled the SDI-induced $O_3$-rich belts in Fig. 4A. It stretched from extratropics (40$^o$N) to subtropics (20$^o$N), approximately 2000 km.

7. **Lines 271-273:** It is difficult for me to see how the wind-driven dispersion of the intruded $O_3$-rich stratospheric air is concluded here. Could you provide more discussion on this?

   Reply: Thank you very much for this comment. We made a wrong explanation for the high $O_3S$ concentrations over the Mongolian Plateau. We adjusted the view to examine the three-dimension structure of $O_3S$, and found a secondary SI emerged over the Mongolian Plateau since June 12. This secondary SI is the reason of high-concentration $O_3$ over the Mongolian Plateau. However, this elevated $O_3$ concentrations had no influence on eastern China due to its downwind location. Lines 241-244, and 284-287.

8. The title, abstract, introduction, and summary all mention "fully-validated EAC4 reanalysis", but the text of results is unclear in explaining this process of "fully-validated". What is the definition of fully-validated?

If it is only reliable O$_3$ correlation validation then being called "fully" is not appropriate even too absolute. It might be that "O$_3$-validated EAC4 reanalysis" or just "validated EAC4 reanalysis" would be sufficient.

Reply: Thanks for this wonderful suggestion. As your previous suggestion, we added a scatter plot to show the well performance of EAC4 reanalysis. We accepted your suggestion and changed the tile to be "Widespread stratospheric intrusion influence on summer ozone pollution over China revealed by multi-site ozonesonde and validated EAC4 reanalysis". In this new title, we deleted "ground-based measurement" because the ground-based measurement is not as important as multi-site ozonesonde and validated EAC4 reanalysis in this study.

**Technical Comment:**

1. **Line 44:** SI event should be plural.

   Reply: Thank you for picking it up. We re-wrote the sentences in the revision.

2. **Lines 49-52:** Could you rearrange the citations? Which references correspond to satellite observations, and which are for atmospheric reanalysis and model simulations?

   Reply: Thank you for this suggestion. We rearranged the citations. Lines 50-54.

3. **Line 53:** Which open-source products? Which custom model simulations? And please include references.

   Reply: The open-source products include the freely-available satellite observations and atmospheric reanalysis. The custom model simulations include the specific numerical simulation for SI event. They are already listed by citations in the previous sentences (see reply to Technical Comment # 2). In the revision, we re-wrote the sentences and deleted some redundant description. Lines 50-56.

4. **Line 56:** Could you name a few chemical tracers?

   Reply: Thank you for this suggestion. In fact, we listed two chemical tracers (isotopic sulfur and O$_3$-CO ratio) in the following sentences. Lines 55-58.

5. **Lines 58-60:** For redibility, I would use "a study using isotopic sulfur ($^{35}$S)" and "a study using O$_3$-CO"

   Reply: Thank you for this suggestion. We re-wrote the sentence as your suggestion. Lines 60-62.

6. **1:** The description in Lines 117-131 reads well. However, it is difficult to find the described details in the figure without zooming in. Could you improve Fig.1, for example, by increasing the font size, highlighting the contours for vorticity, and using a different colour for the wind direction? A good figure should be readable at 100% page view.

Reply: A good suggestion!!! We adjusted the Figure 1 in the revision. We adopted a 4×2 subplot matrix to replace the initial 2×4 matrix. The 4×2 matrix allows for better tracking the westward movement of upper-level trough and synoptic change across time. In the new figure, we characterize the geopotential height and upper-level trough by shading, with trough axis highlighted by red dot lines. Only wind direction of jet exceeding 20 m s⁻¹ was shown by green arrows. The yellow contours were used to highlight the potential vorticity of 1.5 PVU at 400 hPa.

[Figure]

**Fig. 1**. (A) Horizontal distribution of geopotential height (shading, units in gpm), and wind direction of jet exceeding 20 m s⁻¹ (arrows) at 200 hPa, and potential vorticity of 1.5 PVU (yellow contours) at 400 hPa. (B) MODIS satellite cloud images with the dashed box marking eastern China (105°E–

121 E, 21 N–41 N). Red dot lines in (A) denote the axis of upper-level trough at 200 hPa. Magenta circles mark the available ozonesondes at different sites (BJ: Beijing, CC: Changchun, and HK: Hong Kong) on different days.

7. **2:** Could you make the font size and legend larger? Please put the location on the left size of each row. The subplot y-axis title (and scale) at the same row in Fig.2 could share the one because you share x-axis title at the same column. The label font is expected to be appropriately larger, so that the legend (how about a row at the top instead?) might more conducive for reading.

Reply: A good suggestion! As you suggested, we re-plotted the Figure 2. Figure 4 and Figure 5 were also re-plotted.

[Figure]

**Fig. 2**. O$_3$ vertical distribution over (A) Beijing, (B) Changchun, and (C) Hong Kong derived from ozonesonde and other data sources (including AIRS satellite observation, EAC4 and MERRA2 reanalysis) during 10–13 June 2013. Black and blue lines denote the sonde-based temperature (T) and relative humidity (RH) profiles, respectively. Gray dashed lines represent the thermal tropopause height, and gray dot lines indicate the boundary layer top height. Upper-level secondary O$_3$ peaks are shaded heavy gray, and SI-induced O$_3$-rich layer in the troposphere is shaded light gray.

8. **Line 152:** "… sonde-based $O_3$ profiles …" I noticed you used "ozone" instead of single "$O_3$" in this paper, shouldn't here the full name of "ozone" be used?

   Reply: Thank you for picking it up. In the revision, "$O_3$" was used uniformly throughout the manuscript, except when it first appeared.

9. **Lines 196:** I am confused by the use of "filament" and "belt" here.

   Reply: We re-wrote the sentences to avoid confusion. Lines 209-213: "The stratospheric intrusions developed into elongated (about 2000 km) and slender (about 200 km) streamers with elevated $O_3$ concentrations exceeding 150 ppbv (referred to as SDI-induced $O_3$-rich belts) at 400 hPa. On the east of the SDI streamers, the WCB streamers were parallel with anomalously low $O_3$ concentrations (referred to as WCB-related $O_3$-poor belts)". Besides, we labeled the SDI-induced $O_3$-rich belts and WCB-related $O_3$-poor belts in Fig. 4A.

10. **Lines 279-280:** Could you please label the Taihang Mountains and southern NCP on the figure?

    Reply: Thank you for this suggestion. In this revision, we thought it was trivial to highlight the weak $O_3S$ hotspots in the Taihang Mountains and southern NCP. Therefore, we deleted corresponding description in the revised manuscript.

11. **Line 313-314:** "Besides, ground-based stratospheric tracer method had been developed to quantify the stratospheric intrusion contribution over China." How to understand the "ground-based stratospheric tracer method"? "Ground-based" seems to be somewhat of an inconsistency with "" Was it mean ground-based validated stratospheric tracer data? According to the current information, you did not use ground-based data to validate the tracer but only the ozone. Here do need some explanations. Additionally, using "…to quantify the stratospheric intrusion contribution to surface ozone over China" is more explicit.

    Reply: Thank you for this good suggestion. We replaced the "ground-based stratospheric tracer method" by "ground-based chemical tracer method" in the revised manuscript. We had introduced the term of "ground-based chemical tracers" in the Introduction. As you suggested in Technical Comment #4, we also listed some ground-based chemical tracers used in previous studies, e.g., cosmogenic sulfur-35 ($^{35}$S) and the ratio of $O_3$ to CO.